# A Three-Way Synergistic Effect of Work on Employee Well-Being: Human Sustainability Perspective

**DOI:** 10.3390/ijerph192214842

**Published:** 2022-11-11

**Authors:** Sugumar Mariappanadar, Wayne A. Hochwarter

**Affiliations:** 1Peter Faber Business School, Australian Catholic University, Melbourne, VIC 3002, Australia; 2Melvin T. Stith Professor of Business Administration, Department of Management, Florida State University, Tallahassee, FL 32306, USA; 3Honorary Professor, Faculty of Law and Business, Australian Catholic University, Melbourne Campus, Melbourne, VIC 3002, Australia

**Keywords:** health harm, sustainable HRM, well-being, ceiling effect, job tension, supervisory support

## Abstract

We explored the interaction of the United Nation’s sustainable development goals to facilitate human sustainability using occupational health and sustainable HRM perspectives. In Study 1 (*n* = 246), we assessed the preconditions to empirically confirm the distinctiveness of the dimensions of health harm of work from other study constructs. Subsequently, we tested the hypotheses across two studies (*n* = 332, Study 2; *n* = 255, Study 3). In alignment with the ceiling effect of human energy theory, the three-way interaction results across the samples consistently indicate that high supervisory political support (SPS) significantly strengthens the negative interactions of psychological health risk factors and high job tension as adverse working conditions (SDG-8) on working-condition-related well-being as the human sustainability dimension (SDG-3). Similarly, synergistic effects were found of the side effects of work on health, high job tension, and high SPS on well-being in sample 3. We discuss theoretical and future research for human sustainability from occupational health and sustainable HRM perspectives.

## 1. Introduction

United Nations has encouraged organisations to voluntarily adopt sustainable development goals (SDGs) in business strategies to promote human sustainability, which many global companies have embraced. In this context, the perspective of interactions of SDGs for human sustainability at work [1] emphasises that decent (adverse) working conditions (SDG-8) are essential aspects of the health and well-being of the working population (SDG-3). Many definitions exist in the literature for employee well-being [2,3]. However, we chose to use Van Laar, Edwards, and Easton’s [4] definition of employee well-being in this study which relates to the quality of work life or work-related quality of life, to examine the human dimension of sustainability.

The human dimension of sustainability [5,6] uses health harm of work (HHW) as a leading indicator to develop responsible/decent human resource management (HRM) practices (SDG-8) that enhance employee well-being (SDG-3). In particular, the HHW approach examines restrictions imposed by work on employees’ human energy as a resource to achieve positive health and well-being outcomes [7,8]. This view differs from the work recovery experience, which typically emphasises mood regulation and job-stress coping [9].

The human energy discussed in the definition of HHW represents a source of “fuel” necessary for an employee’s work performance and to involve in non-work activities to enhance well-being [10]. When this finite form of human energy lessens due to the cumulative negative effect of working conditions (SDG-8), it is questionable whether organisational support will facilitate subordinates coping strategies to buffer employee well-being (SDG-3). The ceiling effects of human energy help explain this phenomenon. The ceiling effect argues that the human energy trough or limit reached by high levels of human energy depletion resulting from cumulative adverse work experiences inhibits employee well-being [11]. The theory further argues that the employees’ additional organisational resources, such as supervisor support [12] and supervisory political support [13], which facilitates employees’ resource gain cycle, have only a modest effect on employee well-being, if any, beyond the human energy trough.

The ceiling effect of human energy theory extends the Conservation of Resources (COR) theory [14]. The COR principles explain the critical motivation for employee decisions to maintain and foster resources to manage the resource loss cycle during the current work demands. Subsequently, to acquire resources in the resource gain cycle to guard against further depletion in adverse work contexts. According to COR principles, resource loss and gain are independent cycles. However, in extending the COR principles, the ceiling effect highlights the human energy limit or trough between resource loss and gain cycles which explains the variability in employees thriving for work performance and well-being outcomes. That is, the human energy trough between resource depletion and gain cycles must be short to improve the well-being dimension of human sustainability (SDG-3). However, even with personal and organisational motivational sources for resource gain, it is less likely to guard against resource depletion due to cumulative adverse working conditions (SDG-8) to shorten the human energy trough. Shortening the human energy trough between the resource depletion and gain cycles is essential to plan and develop socially responsible HR practices from the sustainable HRM perspective to help employees allow their biological system to naturally reverse itself to gain human energy for future work performance and well-being.

In the health literature, well-being includes health and constitutes a unifying concept (including medical and non-medical priorities) valuable for health improvement [15]. In the management literature, subjective well-being is a broad construct that includes life satisfaction and job satisfaction which reflects positive and negative affectivity or feelings/emotions toward working conditions of jobs [3]. The job quality perspective of well-being is about the extent to which a job has work characteristics and employment-related factors that facilitate favourable or unfavourable feelings for the employee [16]. Following this tradition, we explored the employee well-being dimension of human sustainability from the sustainable HRM perspective using positive and negative affectivity towards working conditions of jobs.

Previous studies in the occupational health and management literature explored using COR theory the buffering role of organisational resources (e.g., supervisor support and supervisor political support) on the relationship between job stress and employee well-being [17]. Evidence-based on COR theory suggests that supervisor support [12] and supervisory political support [13] tend to positively buffer the ‘silo’ or isolated adverse effect of work (e.g., job tension, work stress) and well-being. However, in this study, we use the ceiling effect of the human energy perspective [11] from sustainable HRM to understand the buffering role of organisational support, such as supervisor political support, on the relationship between chronic or ‘cumulative’ harmful working conditions (adverse SDG-8) experienced by employees and employee well-being as a dimension of human sustainability (SDG-3). Supervisor political support (SPS), which is different from supervisory support, focuses on supervisors’ non-sanctioned (political) actions taken to enhance the well-being of one’s subordinates [13]. However, perceived supervisory support [12] explains the support supervisors provide to their subordinates within the scope of a formal supervisory role.

Understanding the synergistic, adverse effects of work characteristics and SPS on organisation and stakeholder (i.e., employees) outcomes from the ceiling effect of human energy viewpoint is essential for the emerging field of sustainable HRM to gain insights into the temporal perspective [18] on the ongoing adverse effects of work on stakeholders. This empirical evidence will help organisations develop strategic corporate social responsibility initiatives to minimise cumulative adverse working conditions imposed on stakeholders (i.e., employees, families, and society) to facilitate SDG-3. Hence, we explore the three-way interactional or synergistic effect of SPS in facilitating HHW—job tension negative complementarity on well-being based on the ceiling effect of the human energy perspective from the occupational health and sustainable HRM literature (see Figure 1).

The three-sample study fills gaps in occupational health, sustainable HRM, and stress literature, from the harm of work and COR perspectives, by demonstrating the bundle or synergistic effects of HHW, job tension, and SPS on employee well-being. Specifically, SPS manifests when a leader exercises subordinate-benefitting political influence to acquire and distribute resources and remove roadblocks impeding contributions and well-being [13]. In terms of execution, the three-study element is essential to replicate the findings to gain greater confidence than single-study designs [19]. First, an essential precondition to testing our hypothesised relationships is empirically confirming the distinctiveness of HHW from other study constructs. Hence, we used Sample 1 to test the measurement model to complement subsequent hypotheses testing. Second, in Samples 2 and 3, study hypotheses were tested to reveal consistency in findings across samples to gain greater confidence than single study designs.

This research makes multiple contributions. First, previous studies in the COR literature have used rest or recovery to manage employee engagement and burnout caused by loss cycles at work [20]. Our study extends the COR literature to enhance our understanding of the human energy trough between resource loss and gain cycles at work due to the synergistic effects of antecedents on the well-being dimension of human sustainability. Thus, it contributes to empirical evidence for the ceiling effect of human energy to address Fritz, Lam, and Spreitzer’s [10] theoretical proposition relating to “how people seek to manage their energy at work” (p. 36) for the benefit of management and business.

Second, the job demand-resources (JD-R) theory of work stress acknowledges the ceiling effect hypothesis of employee engagement for job performance from the individual employee-level perspective [21]. However, we extend research by examining the ceiling effect of human energy from the organisation level in alignment with the institutional perspective of sustainable HRM to explore the interaction between SDG-8 and SDG-3. The ceiling effect maintains that unless organisations implement sustainable HRM characteristics, the human biological system cannot naturally replenish the energy depleted beyond the human energy trough job demands for performance and well-being [22]. This view argues that organisational-level sustainable work-related strategies for SDG-8 are essential to understand the ceiling effect to help employees naturally reverse the human energy depleted at work to improve employee health and well-being (SDG-3). Finally, the study findings will prompt the practical implementation of organisational-level sustainable HRM practices to benefit employees and organisations. Results will facilitate job design with sustainability characteristics to minimise the synergistic effects imposed by work to improve myriad outcomes (i.e., commitment, extra-role behaviour, willingness to remain) in addition to satisfaction and performance.

## 2. Theoretical Background and Hypotheses

### 2.1. Sustainable HRM Theories and Health Harm of Work

Over the last two decades, as an emerging discipline within HRM, sustainable HRM has been defined in several ways, see review by Macke and Genari [23]. We chose to use the synthesis effects perspective to define sustainable HRM because of its relevance to the hypothesised three-way interaction of the study variables. In doing so, we view sustainable HRM as an institutional or organisational response to UN SDGs in line with “HR systems or bundles of HRM practices that engage employees to synthesise increased organisational performance outcome while simultaneously reducing the unsustainable impacts on the natural environment as well as on employees and their families (i.e., stakeholders)” [22]. The chosen definition of sustainable HRM highlights two relevant theoretical perspectives to explain the study variables’ three-way interaction effects.

First, the theory of work harm [8,22] builds on the negative externality of employment practices from the sustainable HRM literature. In particular, the theory explains the unintended harmful health and well-being (SDG-3) impacts on stakeholders (i.e., employees, their families, and society) by the unsustainable internal efficiency-focused high-performance practices (SDG-8), such as work overload and time demand. HHW augments the theory of harm of work about the restrictions imposed by the job on employees as stakeholders. Specifically, this research advocates that employees involved in non-work-related activities (e.g., time for regular physical exercise, work-family balance) improve positive health and well-being outcomes while enhancing organisational performance. For example, employees may work long hours and take on additional workloads to improve their career advancement or avoid being redundant (i.e., lesser job security). These actions ‘restrict’ employees from non-work activities that promote positive health and well-being outcomes.

HHW includes three reflective dimensions highlighting work restrictions’ unique effects on employee health [7,8]. Specifically, the three dimensions include work restrictions on health (WRH), risk factors of work on psychological health (RFPH), and side effects of work for health (SE) [7]. WRH focuses on the restrictions imposed on employees in non-job-related activities to improve their health based on the stimulus attribution of health harm of work. The extent to which work as the stimulus restricts employees from engaging in non-job activities (e.g., weight control initiatives, social activities, physical exercise) related to positive health outcomes.

RFPH expands on the manifestation or leading sign attribution of work harm, emphasising the risks of work restrictions on an employee’s cognitive and behavioural conditions (e.g., emotionally drained) that facilitate negative health issues. Identifying work restrictions for employees helps identify the risk factors that mitigate early to prevent or delay the onset of work-related diseases (e.g., cardiovascular disease, sleeplessness). SE emphasise the causal attribution of HHW, which is about the ‘unintended’ adverse side effects imposed on employees while an organisation pursues its ‘intended’ business goals. For example, work intensification organisations use to achieve intended improved business bottom-line goals imposes unintended adverse effects on employees, such as excessive use (4 to 10 large cups a day) of coffee and alcohol. These unintended adverse effects of HHW are likely to lead to work-related health problems [24].

Finally, the synthesis effect of sustainable HRM indicates that improved organisational performance and simultaneously reducing HHW operate as two mutually reinforcing polarities [25,26]. Thus, organisations are responsible for providing support to reduce HHW to improve employee well-being [27]. Accordingly, the synthesis effect of sustainable HRM facilitated our examination of potential synergistic (adverse) effects of dimensions of HHW, job tension, and SPS on the well-being dimension of human sustainability.

### 2.2. Literature Review on Relationship between Job Tension and Well-Being

HRM policies and practices set job quality expectations for employees to perform [16]. These job quality expectations are the behavioural requirements or limits that employee needs to acknowledge to perform at work [21]. Employees attempting to satisfy the behavioural job requirements experience difficulties that trigger tension. Job tension describes the effect of stressful working conditions that employees have experienced, reflecting an element of the work stress phenomenon [28,29].

Employee well-being is understood in the literature from the context-free [30] perspective (e.g., life satisfaction, happiness) and domain-specific or job-specific perspective [31]. In this study, we focus on domain-specific workplace employee well-being as a construct that includes positive and negative affectivity toward working conditions of jobs [3]. A meta-analysis on employee well-being revealed that specific employee well-being is best understood in the job domain based on positive outcomes such as job satisfaction [32]. Furthermore, a review of precipitating factors indicates that job satisfaction is predominantly understood from the individual and situational/work environmental perspectives [33]. The work environmental perspective is relevant for this study because sustainable HRM is an institutional theory [22] that considers different situational aspects of work (e.g., job demand, role clarity) imposed by organisations on employees as sources for job tension which act as antecedents for job domain specific employee well-being [28]. Hence, we included the job domain-specific well-being dimension of human sustainability at work based on the quality of job perspective to explore the interaction of SDG-8 and SDG-3 [16].

A meta-analysis revealed that experiencing job tension by employees is consequential of the work environment that negatively affects employee well-being [34]. Furthermore, other studies found that high job tension buffered employee well-being [29,35]. Hence, in this study, we explore the moderation effect of job tension on the relationship between HHW and well-being as a dimension of human sustainability (SDG-3).

### 2.3. Health Harm of Work Dimensions—Job Tension Complementarity on Well-Being

The hindrance stressor perspective of the challenge-hindrance stress model [36] maintains that job demands are likely to affect employee job performance and personal goals negatively. These stressors align with understanding stress and stressors as adverse and strain-inducing effects on employee performance. Hence, work stress encompasses the effects of HHW. However, evidence in the sustainable HRM literature [7] suggests that the HHW construct differs from the hindrance stressors perspective-based work recovery experience construct [9].

First, work recovery experience focuses on restoration after a stressful job situation to improve employee health. However, HHW addresses the restrictions imposed by work on employees to involve in non-work activities to achieve positive health and well-being outcomes. Second, HHW expands the positive health perspective, which advocates a proactive approach to identifying work-related psychological disorders and chronic disease indicators to enhance well-being. Hence, the dimensions of HHW are ‘leading’ indicators for employee positive well-being outcomes, instead of the work recovery experience which typically emphasises mood regulation and job-stress coping as ‘lagging’ indicators of work [9].

Sinelnikov et al. [37] explained the difference between leading and lagging indicators in occupational health. Leading indicators provide organisations with proactive opportunities to detect and mitigate the risk factors imposed on employees to prevent work-related illnesses. However, work stressors as lagging indicators highlight the onset of physical damage, negatively impacting employee performance and health. Hence, we examine HHW as an independent variable and job tension as a lagging moderator in developing these study hypotheses.

A meta-analysis on job stress revealed that employees who experienced high job tension are most likely to have reduced levels of well-being [38]. These effects then impair health and well-being. Furthermore, there is evidence for exploring the complementary aspects of internally consistent HRM systems on employee health harm of work [39] and employee health [6]. However, studies have yet to explore the adverse complementary effects of the dimensions of HHW and job tension on well-being. This absence requires attention as the synthesis effects theory of sustainable HRM organisations highlights that firms have a social responsibility to reduce the negative complementary effects of HHW and job tension as adverse working conditions (SDG-8) to improve well-being (SDG-3). Hence, it is necessary to empirically establish the complementary effects of HHW and job tension on well-being as a dimension of human sustainability at work to curtail the harm of work imposed on employees.

The combined adverse effects of HHW and job tension on well-being are explored based on the self-regulatory depletion mechanism of conservation of resources (COR) theory [14]. The self-regulatory depletion mechanism is the feeling of being preoccupied and tired and exerting high resources and energy levels during chronic working conditions [40]. Hence, the following hypotheses highlight the self-regulatory depletion mechanism of the COR theory in explaining the complementary role of the dimensions of HHW and job tension on well-being.

**H1A.** 
*Job tension will moderate the negative interaction effects of the risk factors of work on psychological health (RFPH) on well-being. Specifically, the negative relationship between RFPH and well-being will be stronger when job tension is high versus low.*


**H1B.** 
*The work restrictions on health (WRH) will moderate the negative interaction effects of job tension on well-being. Specifically, the negative relationship between WRH and well-being will be stronger when job tension is high versus low.*


**H1C.** 
*Job tension will moderate the negative interaction effects of the side effects of work for health (SE) on well-being. Specifically, the negative relationship between SE and well-being will be stronger when job tension is high versus low.*


### 2.4. Supervisor Political Support Strengthens the Health Harm of Work Dimensions—Job Tension Complementarity on Well-Being

Contemporary work settings are characterised by ambiguity and uncertainty, self-serving behaviour, and competition for limited resources to manage work demands. In this context, SPS, which is different from supervisory support, focuses on supervisors’ non-sanctioned (political) actions taken to enhance the well-being of one’s subordinates [13]. However, perceived supervisory support [12] explains the support supervisors provide to their subordinates within the scope of a formal supervisory role. SPS was developed based on the sensemaking theory [41], which suggests that subordinates develop coping strategies based on the understanding of political support provided by supervisors as a boundary condition to buffer the fear and anxiety associated with the contemporary high work-intensive settings on employee well-being.

In the organisational politics literature [42], politics is perceived to be divisive, and managers agree that politics is pervasive in organisational life and can benefit subordinates. Evidence suggests that more than just caring about and valuing subordinates’ contributions from the supervisor support perspective [12], supervisor political support facilitates the sensemaking of organisational realities by subordinates for converting negatively perceived work settings into benefits [13]. Extending the sensemaking conceptualisation of supervisory political support in reframing organisational realities by subordinates to the occupational health and sustainable HRM literature, SPS, which facilitates employees’ coping strategy, has limits on human energy to enhance well-being as a dimension of human sustainability.

Job demand-resources theory examines the ceiling effect hypothesis of employee engagement for performance [21]. Furthermore, in contrast to the positive buffering of SPS on the silo or isolated positive effect of organisational politics and well-being [13], we believe SPS may have a negative buffering effect on the relationship between the combined negative effect of HHW and job tension (i.e., effects of stressful work conditions) on well-being from the ceiling effect of human energy perspective (see Figure 1). As explained earlier, the ceiling effect of human energy is about the inadequacy of supervisory support as an organisational resource in extending employee energy trough or limit to subdue the combined negative effects of HHW and job tension on well-being. When employees experience chronic negative work characteristics, their physical energy or fuel limit for organisational performance lessens. In this context, supervisor or organisational support will not assist in thriving or a sense of vitality [43] to improve well-being. To date, no study has explored the ceiling effect of the human energy hypothesis of SPS in subduing the complementary adverse effects of dimensions of HHW and job tension as working conditions (SDG-8) on well-being as a dimension of human sustainability (SDG-3). Hence, we explore the three-way interaction effects of HHW, job tension, and SPS on well-being to address this gap. Hence, we propose the following hypotheses:

**H2A.** 
*The high SPS strengthens the negative interaction effects of risk factors of work on psychological health (RFPH) and job tension on well-being. Specifically, the proposed negative effect in Hypothesis 1A is stronger when high SPS is present.*


**H2B.** 
*The high SPS strengthens the negative interaction effects of work restrictions on health (WRH) and job tension on well-being. Specifically, the proposed negative effect in Hypothesis 1B is stronger when high SPS is present.*


**H2C.** 
*The high SPS strengthens the negative interaction effects of the side effects of work on health (SE) and job tension on well-being. Specifically, the proposed negative effect in Hypothesis 1C is stronger when high SPS is present.*


## 3. Methods

### 3.1. Research Design

A three-sample study design tested the hypothesised three-way interaction on well-being to address method concerns and increase confidence in the findings relative to single-study designs [19].

### 3.2. Data Collection

#### 3.2.1. Sample 1 (Study Model Testing)

Consistent with prior investigations [44], students in a large introductory management course distributed a survey to full-time employees who were given course credit. We gathered the independent variables (i.e., tension, HHW, and SPS) and control variables (i.e., age, gender, organisational tenure) at Time 1 and the dependent variable (i.e., well-being) at Time 2. Approximately 30 days separated the two data collections. Respondents provided email addresses during each time for matching purposes. We received 349 surveys that we reviewed for adequacy using several screening procedures to identify careless responses suggested by Hochwarter et al. [19]. First, we identified participants finishing the survey in less than 2 s per item. Second, we noted participants who failed an attention check. Third, we checked for invariant responses in the focal scales (e.g., tension, HHW, SPS, and satisfaction). We eliminated 12 responses leading to a final sample size of 337. All participants worked in the United States. The sample was 54 percent males (*M* = 1.54, *SD* = 0.50), averaged 37 years of age (*M* = 36.85, *SD* = 14.69), and reported 40 h of work per week (*M* = 40.07, *SD* = 10.45).

#### 3.2.2. Sample 2 (Hypotheses Testing)

We distributed surveys to all 346 employees of a local municipality via email. The municipality resided in the Southeast United States. After two weeks and one prompt, we received 261 surveys (response rate of 80%). However, we used the screening procedures described above, requiring the elimination of 14 surveys. Participants worked in representative job titles (e.g., accountant, customer service, technical support). The sample was 47% males (*M* = 1.47, *SD* = 0.51), approximately 36 years old (*M* = 35.76, *SD* = 14.74), and worked 39 h per week (*M* = 39.24, *SD* = 10.50).

#### 3.2.3. Sample 3 (Hypotheses Testing)

We distributed an information letter to a sample of 320 employees from six companies in Australia, which included an URL to access a web-based questionnaire. A total of 255 usable questionnaires were collected (response rate of 80%). 52% of participants were females (*M* = 1.52, *SD* = 0.50), and 49 percent aged between 26 years and 50 years (*M* = 32.32, *SD* = 13.71). Thirty percent of participants completed an undergraduate degree (*M* = 3.21, *SD* = 1.24), and 56 percent worked more than 40 h per week (*M* = 38.34, *SD* = 10.84).

### 3.3. Measures

We measured all constructs using a scale anchored with 1 (strongly disagree) and 7 (strongly agree) as endpoints.

#### 3.3.1. Job Tension

We measured job tension with House and Rizzo’s [45] five-item scale. “I work under a great deal of tension” is a representative item (Sample 1, α = 0.86; Sample 2, α = 0.87; Sample 3, α = 0.74).

#### 3.3.2. Employee Well-Being

In the occupational health and management literature, there are different measures for employee well-being covering work- and non-work-related characteristics [46,47]. In this study, similar to other studies in the literature, we used employee well-being as a work characteristic related to job domain-specific construct based on the job satisfaction measure [48]. We used Brayfield and Rothe’s [49] index to measure job satisfaction from the job domain-specific perspective. “Most days I am enthusiastic about my work” is a representative item (Sample 1, α = 0.89; Sample 2, α = 0.72; Sample 3, α = 0.78).

#### 3.3.3. Health Harm of Work

The existing scale includes three dimensions: work restrictions on health (WRH), the risk factors of work on psychological health (RFPH), and the side effects of work for health (SE) [7]. The dimension on RFPH includes five items. “My emotional health is negatively affected” is a representative item (Sample 1, α = 0.81; Sample 2, α = 0.82; Sample 3, α = 0.80). The dimension of WRH includes four items. “It is difficult for me to find time to implement strategies to control my weight” is a representative item. (Sample 1, α = 0.77; Sample 2, α = 0.71; Sample 3, α = 0.71). Lastly, the dimension on SE includes four items. “I have felt that my work pressures cause disturbances to normal sleep” is a representative item (Sample 1, α = 0.72; Sample 2, α = 0.83; Sample 3, α = 0.77).

#### 3.3.4. Supervisor Political Support (Sample 1 to 3)

We measured SPS using Kane-Frieder et al.’s [13] five-item scale. “The boss has knocked down many roadblocks that I have faced by manipulating the system”, sample 1, α = 0.86; sample 2, α = 0.87; sample 3, α = 0.74).

#### 3.3.5. Control Variables

A meta-analysis found that employee characteristics, such as increased age, reduced job tension, and increased well-being outcomes [50]. Furthermore, this research reported that women-dominated groups had increased job tension and reduced well-being compared with men-dominated groups when gender bias exists in organisational employment decisions [34]. Similarly, compared with legally prescribed working hours, overworked employees are likely to have increased employee health harm [7]. Hence, we entered age, gender, and working hours as control variables.

## 4. Analytical Approach

We used hierarchical moderated multiple regression analyses to explore the three-way interactions of dimensions of HHW and job tension on well-being (SPSS 27). We performed collinearity diagnostics to evaluate the impact of method variance on study findings. Specifically, we considered variance inflation factors (VIF), which measure the magnitude of collinearity among predictors of a regression model [51]. Furthermore, we conducted tolerance tests (TOL) to provide a complementary collinearity index. VIF scores less than ten are considered satisfactory [52].

## 5. Results

### 5.1. Descriptive Statistics

We present descriptive statistics (mean, SD, and correlations) for Samples 1–3 in Table 1. Hair and colleagues [52] suggested that correlation coefficients above 0.70 indicate potential multicollinearity. None of the correlations were above the threshold in the study.

### 5.2. Measurement Model Findings (Sample 1)

We conducted a series of CFAs to demonstrate the independence of study variables in Sample 1. We used six fit indices [52], including chi-square goodness-of-fit statistics, RMSEA, GFI, CFI, TLI, and IFI, to determine how multiple models fit the data. In addition, we tested a complete measurement model initially using CFA in which all items were loaded onto their respective latent factors [52]. The six-factor model fits well with the data compared with the twelve alternative models (see Table 2). Hence, all variables in the study were distinct and, thus, are included in further analyses.

These include employee well-being (EW); risk factors of work on psychological health (RFPH); work restriction on health (WRH); side effects of work for health (SE); job tension (JT); supervisor political support (SPS).

We show regression results of RFPH, WRH, and SE on well-being for Samples 2 and Sample 3 (see Table 3, Table 4 and Table 5, respectively). As shown, results indicate a significant negative relationship between RFPH and well-being in both Sample 2 (B = −0.11, SE = 0.04, *p* < 0.05) and Sample 3 (B = −0.21, SE = 0.04, *p* < 0.01). Furthermore, there exists a significant negative relationship between WRH and well-being in Sample 3 (B = −0.07, SE = 0.03, *p* < 0.05) but not in Sample 2 (B = −0.03, SE = 0.03, *p* ns). Lastly, there is no significant relationship between SE and well-being in Samples 2 and 3 (see Table 5).

### 5.3. Test of Hypotheses

#### 5.3.1. Complementarity of Dimensions of Health Harm of Work and Job Tension on Well-Being

We report the effects of the interactions of each HHW dimension (e.g., RFPH, WRH, and SE dimension) and job tension on well-being for Samples 2 and 3 (see Table 3, Table 4 and Table 5). In these tables, we also include the interaction effects of HHW dimensions and SPS on well-being. There is no evidence of a significant negative interactive relationship between RFPH and job tension on well-being among both Samples 2 (B = −0.02, *SE* = 0.02, *p ns*) and 3 (B = −0.04, *SE* = 0.03, *p ns*). However, this interaction was significant for WRH (B = −0.06, *SE* = 0.02, *p* < 0.05), SE (B = −0.06, *SE* = 0.03, *p* < 0.05), and job tension on well-being, respectively, only among Sample 3. Hence, we rejected Hypothesis 1A. Conversely, Hypothesis 1B and Hypothesis 1C were partially supported.

#### 5.3.2. Three-Way Synergistic Effects of Dimensions of Health Harm of Work, Job Tension, and Supervisor Political Support on Employee Well-Being

We report the three-way interaction effects of each HHW dimension with job tension and SPS on well-being in Table 3, Table 4 and Table 5, respectively. We report a significant three-way interaction (negative) of RFPH, job tension, and SPS on well-being (see Table 3) in Samples 2 (B = −0.03, *SE* = 0.01, *p* < 0.05) and 3 (B = −0.04, *SE* = 0.03, *p* < 0.05). Further, we report a significant three-way interaction of SE, job tension, and SPS on well-being (see Table 5) only in Sample 3 (B = −0.07, *SE* = 0.03, *p* < 0.01). We found no three-way interaction (see Table 4) of WRH, job tension, and SPS on well-being in both Samples 2 (B = −0.02, SE = 0.01, *p ns*) and 3 (B = −0.01, *SE* = 0.01, *p ns*). Hence, Hypotheses 2A and 2C were accepted, while Hypothesis 2B was rejected. Lastly, regressions revealed that variance inflation factors were below 2.0, highlighting that multicollinearity was not overly problematic in the study.

We used a method suggested by Dawson and Richter [53] to determine if the slopes of each HHW dimension on well-being differed in form and magnitude under various conditions (i.e., high and low conditions of job tension and supervisor political support). To confirm Hypothesis 2A, there must be negative effects of RFPH on well-being under high job tension and high SPS conditions. Moreover, these adverse effects should be more significantly negative than the effects of low job tension and low SPS. We show the three-way synergistic effects findings in Figure 2 for Sample 2 and Figure 3 for Sample 3. The simple slope values and differences in these values are shown in Table 6.

The slope of the relationship between RFPH and well-being is negative and significant in Samples 2, and 3 for the “High job tension—High supervisor political support” condition (difference of −0.87 and −1.07 for Sample 2 and Sample 3, respectively) Line (1) in Figure 2 (well-being—Sample 2) and Figure 3 (well-being—Sample 3) show the more significant negative relationship. In addition, the slope differences between “High job tension—High SPS” and “Low job tension—Low SPS” conditions are negative and significant for both Sample 2 and Sample 3 (difference of −0.60 and −0.71 for Samples 2 and 3, respectively) when the job tension is high. This finding indicates that the interaction effect of RFPH and job tension on well-being is significant in the negative direction of high SPS.

The slope of the relationship between SE and well-being is negative and significant only in the “High job tension—High SPS” condition for Sample 3 (difference of −0.26). Line (1) in Figure 4 (well-being) for Sample 3 shows a more significant negative relationship. The slope difference between “High job tension—High SPS” and “Low job tension—Low SPS” conditions is significant (difference of −0.17, see Table 7). This result indicates that the interaction effect of SE and well-being is significant in the negative direction of high SPS. The findings demonstrate that the synergistic effects of RFPH and SE and job tension strengthened significantly in a negative direction on well-being when considering high SPS.

## 6. Discussion

We conducted a three-sample study to investigate the three-way interaction or synergistic effect of HHW (PFPH, WRH, and SE), job tension, and SPS as an organisational resource on well-being as a work-related well-being dimension of human sustainability. Our study builds upon the earlier conceptualisation of all study variables to establish that these content domains are separate and distinct by using sample 1. A previous study from the sustainable HRM literature [7] found that HHW and work recovery experience [9] are two distinct constructs. The multiple confirmatory factor analysis in testing the study model (Figure 1) using sample 1 extends the theory of harm of work from the sustainable HRM perspective demonstrating the benefit of considering HHW and job tension as two distinct constructs to understand the complementary effects of these variables on well-being.

The study findings extend empirical evidence of the complementary or cumulative effect of HHW and job tension on well-being as work characteristics for well-being. Previous studies [54] explored the cumulative effects of job demand dimensions and job control on employee exhaustion and vigour (i.e., energy and enthusiasm) over time. They found that stable high job demand overtime contributes to increased exhaustion, and the group with increasing job control overtime experienced reduced exhaustion and increased vigour. Although in our study, the WRF and SE dimensions of HHW moderated the negative relationship between job tension and well-being in Samples 2 and 3, it was significant only in sample 3. Previous studies in occupational health and sustainable HRM literature revealed a direct relationship between high-performance work practices [26,55], work intensification [39], and HHW. This study broadens empirical evidence from the occupational health and sustainable HRM perspectives by establishing the interaction effects of high job tension as adverse working conditions (SDG8) in facilitating the negative relationship between dimensions of HHW (i.e., WRH and SE) and employee well-being (SDG3) for the human dimension of sustainability at work.

There is no evidence of the envisaged negative complementarity effects between RFPH and job tension on well-being among both samples. Research methods on moderation study indicate that when there is a weak link between the independent variable and the outcome variable in the presence of a moderator, it is helpful to explore with an alternative/additional moderator or mediated moderation [56]. Hence, the three-way interaction analysis in this study found that RFPH, high job tension, and high SPS curtail well-being, and it was consistent across Sample 2 and Sample 3. Similarly, the three-way interactions revealed that high SPS has a significant negative complementary effect of job tension and SE on well-being in sample 3. Still, the three-way interaction effect remained negative in sample 2 while not significant. These findings support the ceiling effect from the sustainable HRM perspective. The findings underscore the inadequacy of high SPS to help employees strive for resource gain to manage physical energy depleted during cumulative negative events of adverse working conditions (SDG-8) to buffer employee well-being (SDG-3).

### 6.1. Theoretical Contributions

This study is the first in the occupational health, sustainable HRM, and work stress literature exploring the complementarity of dimensions of HHW and job tension on well-being hypotheses. Our study provides empirical evidence to extend the attribution theory of harm of work, COR theory, and the ceiling effect of human energy theory from sustainable HRM for understanding the complementary adverse effects of HHW and job tension on well-being to plan organisational-level interventions for SDG-8 and SDG-3 to enhance human sustainability.

First, based on the attribution theory in the JD-R model, the strength of association between the attribution of the independent variable (i.e., work stressors) and outcome variables (e.g., well-being, employee turnover) was explained as a function of moderators [57]. Similarly, the study findings contribute to the attribution of harm of work theory [7]. As discussed earlier in the background section, the WRH, RFPF, and SE dimensions of HHW, used as independent variables, were developed based on stimulus-source attribution, manifestation or leading sign attribution, and causal attribution of the harm of work theory, respectively.

The study findings provide empirical evidence to the stimulus-source attribution perspective of harm of work theory in explaining that the function of WRH is negatively related to well-being when high job tension exists at work. When work restricts employees from being socially and physically reinvigorating activities for positive health, as a stimulus source for harm of work, it complements the negative effect of high job tension on employee well-being. Drawing from the causal attribution aspect of harm of work theory, the findings contribute to occupational health and sustainable HRM literature that the negative relationship between SE and well-being was buffered by high job tension at work. For example, work’s unintended adverse consequences or side effects on employee health, imposed by work practices while attempting to achieve organisational goals, negatively buffer well-being when high job tension is present.

Second, our study provides ample empirical evidence for understanding the complementary negative effects of each of the two dimensions of HHW (i.e., WRPF and SE) and high job tension on well-being. Furthermore, outlining the self-regulatory depletion mechanism of COR theory [40], we explain that employees feeling of preoccupied and tired after a high level of human energy is depleted due to the complementary negative effect of each of the two dimensions of HHW (i.e., WRH and SE) and high job tension will buffer well-being. The intensity of the complementary negative effect of WRH and SE harm of work and high job tension was higher than the silo or isolated effect of these variables in curtailing employee well-being based on well-being.

Third, studies in JD-R and JD-C from the occupational health literature explore the three-way interaction effect of internal resources, job control, and job stressors on employee well-being and well-being [58]. However, the synergistic or three-way interaction effect of HHW, job tension, and SPS on employee well-being is rare in the literature on occupational health, sustainable HRM, SDG, and job strain. Hence, to indicate how our study extends empirical evidence, we discuss previous studies about the two-way interaction effect of organisational resources, such as perceived organisational support [59] and SPS [13,60], found to improve well-being during job tension experienced by employees. Similarly, the synthesis perspective of sustainable HRM studies on the two-way interaction of perceived organisational support buffers the negative effect of high-performance work practices on HHW [27,36,55].

In expanding empirical evidence on previous three-way interaction and two-way interaction studies in the occupational health, job strain, and sustainable HRM literature, the current study suggests that high SPS does not represent the resource gain cycle of COR theory to buffer the negative complementarity between the two dimensions of HHW (RFPH and SE) and job tension on well-being. Hence, the findings extend the COR theory with empirical evidence for the ceiling effect of human energy theory from the occupational health and sustainable HRM perspectives [11].

Fourth, the ceiling effect findings extend the temporal aspect [18] of the cumulative adverse effects of working conditions on employee well-being for our improved understanding of human sustainability from sustainable HRM. Even with high SPS for employee resource gain, it is less likely to guard against resource depletion over a period due to cumulative adverse working conditions (SDG-8) to shorten the human energy trough. Shortening human energy trough between the resource depletion and gain cycles is vital to help employees allow their biological system to naturally reverse itself to gain human energy for future work performance and well-being. Hence, the findings will enable organisations to plan and develop socially responsible HR practices from the sustainable HRM perspective to facilitate employee well-being (SDG-3). Finally, this study provides empirical evidence for the ceiling effect of human energy to address Fritz, Lam, and Spreitzer’s [10] theoretical proposition relating to “how people seek to manage their energy at work” (p. 36) for the benefit of management and business.

### 6.2. Limitations and Future Directions

Although this study has many strengths, including replicating and extending findings on the synergistic effect of study variables on well-being across three unique samples, it is not without limitations. We used convenience sampling to identify full-time employees for the three different samples in the study. Future studies should benefit from attempting to identify a representative sample of the larger group of full-time employee characteristics based on gender, age groups, and industry types. Despite controlling for similar constructs (i.e., HHW and job tension), and testing multiple models by conducting CFA to demonstrate the independence of study variables to explore well-being, furthering this effort will facilitate better conceptualising and developing theoretical space for organisations to understand subjective employee well-being to improve human sustainability.

A single cross-sectional, self-report data collection introduces common method bias (CMS) into the results. We used several procedural remedies indicated by Podsakoff et al. [61] to mitigate the effects of CMB. For example, there is evidence that CMB does not create artificial interaction effects [62]. Hence, CMB was not likely to affect the interaction effects reported in our findings. Finally, the hypotheses tested in this study used a multi-sample design to replicate the findings with a higher confidence level than the single-study designs [19].

Future research would benefit from a longitudinal design to explore the dynamics of complementary and synergistic effects of HHW along with other employment practices (i.e., job design, work intensification) from the temporal perspective to understand the ceiling effect of human energy at work on job performance, employee engagement and human sustainability outcomes (i.e., employee health and well-being). This study’s results failed to support the envisaged synergistic effect of WRH on well-being. Hence, future research can also explore the indirect effect of prosocial job design on the cumulative conditional level of negative interactional effects of HHW and job tension using a mediated moderation study design [63].

### 6.3. Practical Implications

Our findings promote sustainable HRM practices as an institutional or organisational response to the UN’s SDG-8 and SDG-3. Work intensification has become a dominant source that facilitates harm to work and job tension in increased absenteeism, presenteeism, and employee turnover and creates a competitive disadvantage to organisations and social costs to stakeholders [6,25]. Hence, the study results indicate that practitioners must note the complementary effect of HHW and job tension in reducing employee well-being, leading to a loss of human capital and competitive advantage. Hence, practitioners must re-think their management practices on SDG-8 (decent working conditions) while attempting to reduce the single negative effect of work that will improve well-being as a dimension of human sustainability (SDG-3).

In the occupational health and sustainable HRM literature, organisations cannot avoid imposing social costs of harm of work on employees while the organisation focuses on improved performance (i.e., profit). However, organisations also have a corporate social responsibility to minimise such harm of work imposed by adverse working conditions (SDG-8) on employee well-being (SDG-3). Practitioners know from the JD-R literature that adequate organisation resources will support employees to buffer well-being when employees experience high job tension [60]. However, the synergistic effect results based on the ceiling effect of human energy theory highlight to practitioners that organisational resources provided to support employees to manage their depleted physical energy during cumulative negative events and adverse working conditions will have a limited buffering effect on employee well-being to retain human capital for competitive advantage. Furthermore, the findings revealed that it is essential for the board of companies and operational managers to become aware of the cumulative effect of HHW and job tension as adverse working conditions (SDG-8) to understand the ceiling effect of human energy to re-design jobs/roles to help employees naturally reverse the human energy depleted at work to improve employee health and well-being (SDG-3).

Finally, a literature review article revealed for practitioners that improvements in well-being and job performance are associated with a bundle of employment practices that includes job design which focuses on employee welfare [2]. Hence, practitioners must consider re-designing jobs and roles with prosocial sustainability characteristics to facilitate decent working conditions (SDG-8). Subsequently, that will minimise the source of harmful effects of job stressors and harm of work in reducing employee well-being as a dimension (SDG-3) for human sustainability [64]. For example, practitioners can facilitate employees to craft their job with prosocial sustainability characteristics, including motivational characteristics [65], and be conscientious in minimising the negative effects of long-term exposure to tasks with extensive challenges and responsibilities for job holders.

## 7. Conclusions

This three-sample study facilitates our understanding of the adverse synergistic effects of HHW, job tension, and SPS on employee well-being as a dimension of human sustainability. Our findings provided evidence for the ceiling effect of human energy when SPS as an organisational resource was inadequate to buffer the adverse complementarity effect of the dimensions of HHW (RFPH and SE) and high job tension as working conditions (SDG-8) on employee well-being as a human sustainability dimension (SDG-3). This study meaningfully extends the occupational health, sustainable HRM, and COR literature that human energy at work is not infinite when employees encounter cumulative adverse working conditions. Hence, it is crucial to examine the bundle of negative effects instead of attempting to manage a single negative effect of work in understanding the sustainability of human energy to enhance employee well-being as a dimension of human sustainability and subsequently benefit organisations.

## Figures and Tables

**Figure 1 ijerph-19-14842-f001:**
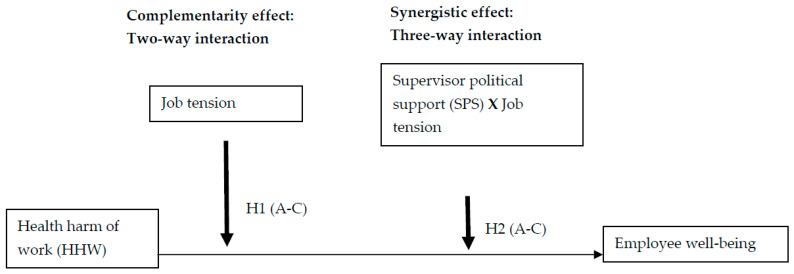
Complementarity and synergistic effects of health harm of work, job tension, and supervisor political support on employee well-being.

**Figure 2 ijerph-19-14842-f002:**
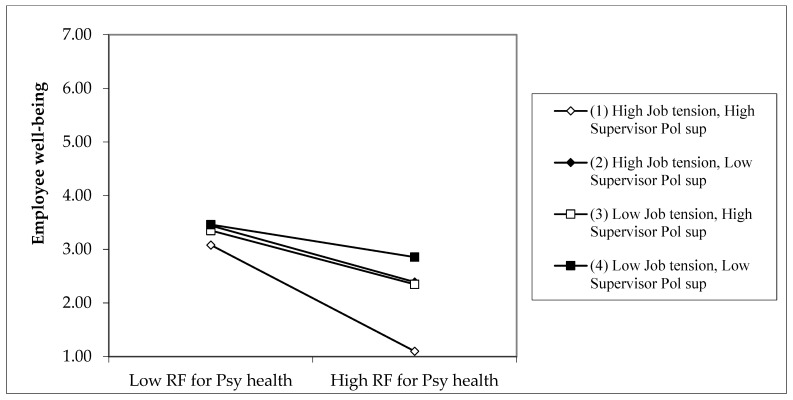
Sample 2: Three-way interaction of risk factors of work on psychological health, job tension, and supervisor political support predicting employee well-being. *Note: Calculations are based on coefficients from Table 3*.

**Figure 3 ijerph-19-14842-f003:**
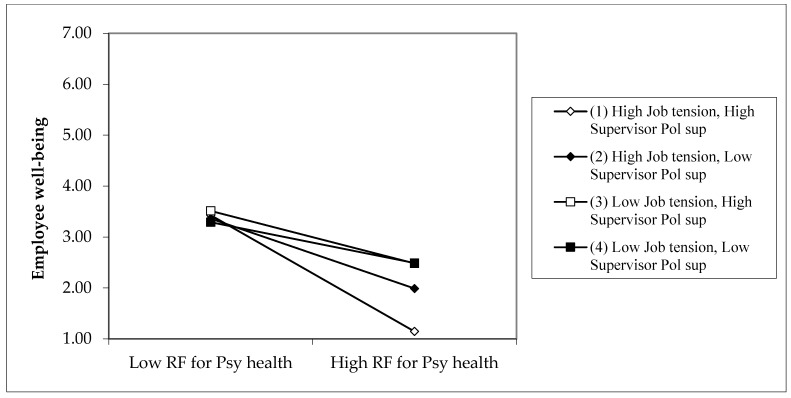
Sample 3: Three-way interaction of risk factors of work on psychological health, job tension, and supervisor political support predicting employee well-being. *Note: Calculations are based on coefficients from Table 3*.

**Figure 4 ijerph-19-14842-f004:**
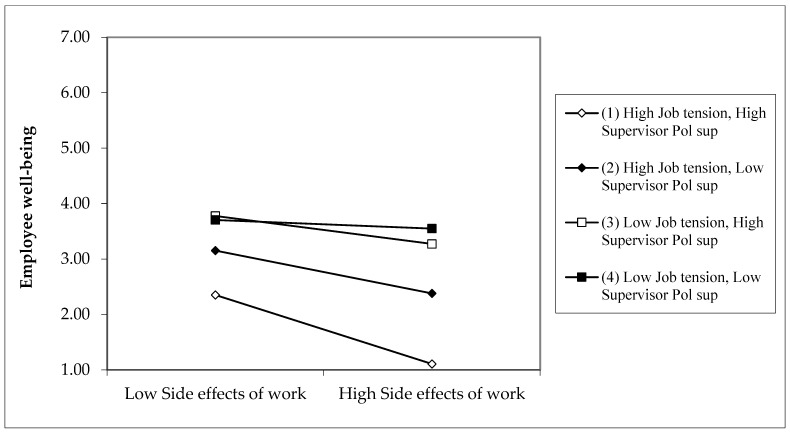
Sample 3: Three-way interaction of side effects of work on health, job tension, and supervisor political support predicting employee well-being. *Note: Calculations are based on coefficients from Table 5*.

**Table 1 ijerph-19-14842-t001:** Descriptive statistics and correlations of in-study variables.

Variable	Mean	SD	1	2	3	4	5	6	7	8	9
1. Age in years (Sample 1)	36.85	14.69	-								
(Sample 2)	35.76	14.74	-								
(Sample 3)	32.32	13.71	-								
2. Gender	1.50	0.52	−0.04	-							
	1.47	0.51	0.09	-							
	1.52	0.50	−0.02	-							
3. Hours per week	40.07	10.45	0.36 **	0.19 **	-						
	39.24	10.50	0.37 **	0.15	-						
	38.34	10.84	0.33 **	−0.16 **	-						
4. Employee well-being (EW)	4.33	0.64	0.07	−0.06	0.06	-					
	4.29	0.75	0.12 *	0.13 *	0.10	-					
	4.16	0.67	0.10	−0.01	0.09	-					
5. Risk factors of work on psychological health (RFPH)	3.04	1.36	−0.04	0.01	0.04	−0.20 **	-				
	2.94	1.26	−0.18 **	.05	−0.07	−0.13 *	-				
	2.88	1.24	0.06	0.14 *	−0.02	−0.12 *	-				
6. Work restriction on health	3.28	1.52	0.09	0.10	0.10	0.13 **	0.57 **	-			
	3.42	1.42	−0.06	0.19 **	−0.01	−0.03	0.67 **	-			
	3.19	1.42	0.01	0.18 **	0.08	0.11 *	0.65 **	-			
7. Side effects of work for health	3.29	1.46	−0.01	.07	0.05	−0.12 *	0.67 **	0.58 **	-		
	3.34	1.41	−0.19 **	0.02	−0.03	−0.11 *	0.69 **	0.61 **	-		
	3.59	1.32	−0.09	0.05	0.03	0.12 *	0.48 **	0.47 **	-		
8. Job tension	3.78	1.39	0.09	0.08	0.24**	−0.07	0.37 **	0.41 **	0.33 **	-	
	3.67	1.44	−0.10	0.05	0.20**	0.06	0.47 **	0.46 **	0.41 **	-	
	3.16	1.30	0.08	0.01	0.15*	−0.01	0.59 **	0.52 **	0.32 **	-	
9. Supervisor political support	4.01	1.40	0.04	0.05	0.01	0.09 *	−0.05	−0.04	−0.03	0.04	-
	3.17	1.47	−0.01	−0.05	0.10	0.08	−0.13 *	−0.03	−0.13 *	−0.06	-
	2.57	1.32	−0.02	−0.04	0.16 **	0.29 **	−0.09	0.01	−0.02	0.01	-

* *p* < 0.05; ** *p* < 0.01 (*n* = 337 Sample 1; *n* = 332 Sample 2; *n* = 255 Sample 3).

**Table 2 ijerph-19-14842-t002:** Measurement model results.

Model Type	Fit Statistics				
Sample 1	χ^2^ (*df*)	CFI	TLI	IFI	RMSEA
Full hypothesised measurement model, six factors	440 (288)	0.96	0.95	0.95	0.05
Model A, five factors (RFPH and SE combined into a single factor)	542 (293)	0.92	0.91	0.92	0.06
Model B, five factors (WRH and SE combined into a single factor)	513 (293)	0.93	0.92	0.93	0.06
Model C, five factors (RFPH and WRH combined into a single factor)	455 (293)	0.93	0.92	00.91	0.06
Model D, five factors (EW and SPS combined into a single factor)	1173 (293)	0.73	0.68	0.74	0.11
Model E, five factors (JT and SPS combined into a single factor)	998 (283)	0.79	0.74	0.79	0.09
Model F, five factors (EW and JT combined into a single factor)	700 (293)	0.88	0.85	0.88	0.09
Model G. four factors (RFPH, WRH, and SE combined into a single factor)	549 (297)	0.92	0.91	0.92	0.06
Model H, three factors (RFPH, WRH, SE, and JT combined into a single factor)	743 (300)	0.87	0.84	0.87	0.08
Model I, three factors (RFPH, WRH, SE, and EW combined into a single factor)	1162 (300)	0.74	0.70	0.74	0.11
Model J, three factors (RFPH, WRH, SE, and SPS combined into a single factor)	1106 (300)	0.76	0.71	0.76	0.10
Model K, two factors (RFPH, WRH, SE, SPS, and EW combined into a single factor)	1366 (302)	0.67	0.63	0.68	0.12
Model L, one factor (all six factors are combined into a single factor)	2065 (303)	0.47	0.38	0.47	0.15

*n* = 246 Sample 1.

**Table 3 ijerph-19-14842-t003:** Results of three-way interaction of risk factors of work on the psychological health of work predicting employee well-being.

Variable	Employee Well-Being (EW)	
	Sample 2	Sample 3
*Constant*	3.62 *** (0.21)	3.42 *** (0.22)
*Controls*		
Age	0.01 (0.01)	0.05 (0.04)
Gender	0.20 ** (0.08)	0.01 (0.08)
Hours worked per week	0.01 (0.01)	0.01 0(0.01)
*Independent variables*		
Risk factors of work on psychological health (RFPH)	−0.11 ** (0.04)	−0.21 ** (0.04)
*Moderator*		
Job tension (JT)	0.08 * (0.03)	0.12 ** (0.04)
Supervisor political support (SPS)	0.03 (0.03)	0.06 * (0.04)
*Interaction effect*		
RFPH * JT	−0.02 (0.02)	−0.04 (0.03)
RFPH * SPS	0.001 (0.02)	0.03 (0.03)
JT * SPS	−0.01 (0.02)	−0.04 (0.03)
RFPH * JT * SPS	−0.03 * (0.01)	−0.04 * (0.03)
Δ *R*^2^	0.03 *	0.05 **

* *p* < 0.05.; ** *p* < 0.01.; *** *p* < 0.001; *SE*s are shown in parentheses. *n*= 332 Sample 2; *n* = 255 Sample 3.

**Table 4 ijerph-19-14842-t004:** Results of three-way interaction of work restriction on health predicting employee well-being.

Variable	Employee Well-Being (EW)	
	Sample 2	Sample 3
*Constant*	3.40 *** (0.25)	3.47 *** (0.22)
*Controls*		
Age	0.01 (0.01)	0.07 (0.04)
Gender	0.24 ** (0.09)	−0.02 (0.08)
Hours worked per week	0.01 (0.01)	0.01 (0.01)
*Independent variables*		
Work restrictions on health (WRH)	−0.03 (0.03)	−0.07 * (0.03)
*Moderator*		
Job tension (JT)	0.04 (0.03)	−0.06 (0.04)
Supervisor political support (SPS)	0.03 (0.03)	0.15 *** (0.03)
*Interaction effect*		
WRH * JT	−0.01 (0.02)	−0.06 ** (0.02)
WRH * SPS	0.02 (0.02)	0.02 (0.02)
WRH * SPS	0.01 (0.02)	−0.01 (.03)
WRH * JT * SPS	−0.02 (0.01)	−0.01 (0.01)
Δ *R*^2^	0.02	0.03

* *p* < 0.05.; ** *p* < 0.01.; *** *p* < 0.001; *SE*s are shown in parentheses. *n*= 332 Sample 2; *n* = 255 Sample 3.

**Table 5 ijerph-19-14842-t005:** Results of three-way interaction of side effects of work on health predicting employee well-being.

Variable	Employee Well-Being (EW)	
	Sample 2	Sample 3
*Constant*	3.46 *** (0.27)	3.57 *** (0.25)
*Controls*		
Age	0.01 (0.01)	0.06 (0.04)
Gender	0.22 ** (0.09)	0.01 (0.08)
Hours worked per week	0.01 (0.01)	0.01 (0.01)
*Independent variables*		
Side effects of work for health (SE)	−0.03 (0.03)	−0.05 (0.04)
*Moderator*		
Job tension (JT)	0.04 (0.03)	−0.01 (0.03)
Supervisor political support (SPS)	0.04 (0.03)	0.15 *** (0.03)
*Interaction effect*		
SE * JT	−0.03 (0.02)	−0.06 * (0.03)
SE * SPS	0.02 (0.02)	0.07 ** (0.03)
JT * SPS	0.02 (0.02)	−0.02 (0.02)
SE * JT * SPS	−0.01 (0.01)	−0.07 ** (0.03)
Δ *R*^2^	0.01	0.05 **

* *p* < 0.05.; ** *p* < 0.01.; *** *p* < 0.001; *SE*s are shown in parentheses. *n*= 332 Sample 2; *n* = 255 Sample 3.

**Table 6 ijerph-19-14842-t006:** Slope difference tests of three-way interaction of risk factors of work on psychological health, job tension, and SPS predicting employee well-being. *Note: From Figure 2 and Figure 3*.

Row	Simple Slopes and Their Differences	High Job Tension: +1SD above Mean	
		Sample 2	Sample 3
Row 1	Simple slope for the condition “High job tension (JT)—High supervisor political support (SPS)”: Lines (1) in Figure 2 and Figure 3	−0.87 *	−1.07 ***
Row 2	Simple slope for the condition “High job tension—Low supervisor political support”: Lines (2) in Figure 2 and Figure 3	−0.51 *	−0.70 ***
Row 3	Simple slope for the condition “Low job tension—High supervisor political support”: Lines (3) in Figure 2 and Figure 3	−0.44 *	0.45 **
Row 4	Simple slope for the condition “Low job tension—Low supervisor political support”: Lines (4) in Figure 2 and Figure 3	−0.28 *	−0.36 ***
Row 5	Slope difference between “High JT—High SPS” and “High JT—Low SPS”: Between lines (1) and (2) in Figure 2 and Figure 3	−0.37	−0.37 *
Row 6	Slope difference between “High JT—High SPS” and “Low JT—High SPS”: Between lines (1) and (3) in Figure 2 and Figure 3	−0.44 *	−0.61 ***
Row 7	Slope difference between “High JT—High SPS” and “Low JT—Low SPS”: Between lines (1) and (4) in Figure 2 and Figure 3	−0.60 *	−0.71 **
Row 8	Slope difference between “High JT—Low SPS” and “Low JT—High SPS”: Between lines (2) and (3) in Figure 2 and Figure 3	−0.07	−0.24 *
Row 9	Slope difference between “High JT—Low SPS” and “Low JT—Low SPS”: Between lines (2) and (4) in Figure 2 and Figure 3	−0.23 *	−0.34 **
Row 10	Slope difference between “Low JT—High SPS” and “Low JT—Low SPS”: Between lines (3) and (4) in Figure 2 and Figure 3	−0.16	−0.09

* *p* < 0.05.; ** *p* < 0.01.; *** *p* < 0.001; *n* = 332 Sample 2: *n* = 255 Sample 3. *Note: Calculations are based on coefficients from Table 3*.

**Table 7 ijerph-19-14842-t007:** Slope difference tests of three-way interaction of side effects of work on health, job tension, and SPS predicting employee well-being. *Note: From Figure 4*.

Row	Simple Slopes and Their Differences	High Job Tension: +1SD above Mean
		Sample 3
Row 1	Simple slope for the condition “High job tension (JT)—High supervisor political support (SPS)”: Lines (1) in Figure 4	−0.26 **
Row 2	Simple slope for the condition “High job tension—Low supervisor political support”: Lines (2) in Figure 4	−0.21
Row 3	Simple slope for the condition “Low job tension—High supervisor political support”: Lines (3) in Figure 4	−0.08
Row 4	Simple slope for the condition “Low job tension—Low supervisor political support”: Lines (4) in Figure 4	−0.07
Row 5	Slope difference between “High JT—High SPS” and “High JT—Low SPS”: Between lines (1) and (2) in Figure 4	−0.06
Row 6	Slope difference between “High JT—High SPS” and “Low JT—High SPS”: Between lines (1) and (3) in Figure 4	−0.12
Row 7	Slope difference between “High JT—High SPS” and “Low JT—Low SPS”: Between lines (1) and (4) in Figure 4	−0.17 *
Row 8	Slope difference between “High JT—Low SPS” and “Low JT—High SPS”: Between lines (2) and (3) in Figure 4	−0.13
Row 9	Slope difference between “High JT—Low SPS” and “Low JT—Low SPS”: Between lines (2) and (4) in Figure 4	−0.12
Row 10	Slope difference between “Low JT—High SPS” and “Low JT—Low SPS”: Between lines (3) and (4) in Figure 4	−0.01

* *p* < 0.05.; ** *p* < 0.01; *n* = 236 Sample 3. *Note: Calculations are based on coefficients from Table 5*.

## Data Availability

Data for the study will be provided on request.

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
