# Peer review of "A Three-Way Synergistic Effect of Work on Employee Well-Being: Human Sustainability Perspective"

_ijerph, 2022, doi:10.3390/ijerph192214842_

Round 1

Reviewer 1 Report

A Three-way Synergistic Effect of Work on Employee Well-  being: Human Sustainability Perspective

First. The idea of ​​the study seems interesting, different and even necessary.

The abstract is fine   followed  the solid scientific work structure

Introduction

State the study's novel contributions at the end of the introduction.

Literature Review

            Fine  

Methods

Does your sample represent the population? Make sure whether your sample can represent the population. You need to justify.

results 

is clear

Discussion

1.The discussions are somewhat broad and general. I can’t see a clear explanation behind this argument (explore the interaction of United Nation’s sustainable development goals to facilitate  human sustainability at work using occupational health and sustainable HRM perspectives)

2. There is some conclusions information on this part. You should put them in conclusion part .

The conclusions.

Theoretical implication and practical  implication of the research. Better to be under this part

Good luck 

Author Response

Title: A Three-way Synergistic Effect of Work on Employee Well-being: Human Sustainability Perspective

GENERAL COMMENTS TO THE REVIEWERS:

We are excited about the opportunity given to us to revise this manuscript. We have carefully considered the two reviewers’ comments and have provided a detailed account of how we revised the paper based on the comments and recommendations received. We would like to extend our sincere appreciation to the reviewers for their time and effort to provide such insightful comments to improve the quality of the manuscript.

RESPONSE TO REVIEWER 1 COMMENTS:

Recommendation: Minor Revision

Comment 1: The idea of ​​the study seems interesting, different and even necessary.

Response: Thanks

Comment 2: The abstract is fine followed the solid scientific work structure.

Response: Thanks

Comment 3: State the study's novel contributions at the end of the introduction.

Response: Yes, the part on the contributions to the literature is now moved to the end of the introduction section.

Comment 4: Literature Review - Fine 

Response: Thanks

Comment 5: Does your sample represent the population? Make sure whether your sample can represent the population. You need to justify.

Response: Based on this suggestion, we have now revised the Limitations and Future Direction section in the manuscript.

Comment 6: Results – clear

Response: Thanks

Comment 7: The discussions are somewhat broad and general. I can’t see a clear explanation behind this argument (explore the interaction of United Nation’s sustainable development goals to facilitate human sustainability at work using occupational health and sustainable HRM perspectives)

Response: We have now revised a part of the discussion section (page 19, line 582 – 586), and also the practical implications sub-section (page 21, line 7223 – 728) to address this comment.

Comment 8: There is some conclusions information on this part. You should put them in conclusion part.

Response: No action was taken on this comment because it is not clear in identifying the specific part in the Discussion section which the reviewer is referring to.

Comment 9: The conclusions - Theoretical implication and practical implication of the research. Better to be under this part.

Response: We appreciate your suggestion, but it is a common journal format to include theoretical and practical implications in the Discussion section. Hence, no action was taken on this comment.

Reviewer 2 Report

There is much to admire about this paper – the care with which measures were chosen and tested (sample 1) and the use of multiple samples to test the hypotheses. However, I do feel some improvements are required, as detailed below.

The paper needs thorough overhaul in terms of language and much sleeker presentation is required – I found myself having to read some passages multiple times to understand them.

The hypotheses also need a lot more development. The introduction introduces a lot of concepts, is repetitive, but doesn’t explain in any detail the hypotheses. H2a to H2c in particular could be argued the other way around and make more sense the other way around - that is supervisor support lessens the combined effects of job tension and risk factors. Indeed you state “In this context, supervisor or organizational support will not assist in thriving or a sense of vitality (Kleine, Rudolph & Zacher, 2018) to improve well-being” (p 7 line 331-333). If this is the case, then the combined effect of tension and risk factors etc will be the same regardless of the level of supervisor support. But this is not what you state in the hypotheses and not what you test. Therefore, you need a clearer and more detailed explanation of these hypotheses. It seems to me that supervisor political support can have negative as well as positive connotations and therefore could be (in JDR terms) as resource in some circumstances but a demand in others. This line of reasoning needs to be drawn out a lot more.

There are also too many acronyms. For example ‘risk factors of work on psychological health (RFPH)’. It would be easier for the reader to use simpler terms.

Relatedly, I’m not sure that the term ‘silo effect’ used in the paper in a few places is a term that would be familiar to readers and I wonder whether it refers to something else (e.g., main effect) – but I couldn’t be sure.

There are also needs to be a clear definition of the term ‘ceiling effect’ early in the paper.

P 2 line 84 ‘positive and negative affectivity towards working conditions of jobs’ – do you really mean trait affect – which is what ‘affectivity’ usually refers to

P 3 line 173-176 ‘HHW augments the theory of harm of work about the restrictions imposed by the job on employees as stakeholders. Specifically, this research advocates that employees involved in non-work-related activities improve positive health and well-being outcomes while enhancing organizational performance’ Do you mean any non-work-related activity? Surely only positive ones (which will need to be defined on why they are positive)

I’m not sure section 2.2 is needed at all.

Pp 10 line 429. Correlations > .70. Table 1 indicates two correlations above .70. Could we have some clarification please?

P 14, line 495-496 – surely H2c is only partially accepted given it was found in only one sample.

Author Response

Title: A Three-way Synergistic Effect of Work on Employee Well-being: Human Sustainability Perspective

GENERAL COMMENTS TO THE REVIEWERS:

We are excited about the opportunity given to us to revise this manuscript. We have carefully considered the two reviewers’ comments and have provided a detailed account of how we revised the paper based on the comments and recommendations received. We would like to extend our sincere appreciation to the reviewers for their time and effort to provide such insightful comments to improve the quality of the manuscript.

RESPONSE TO REVIEWER 2 COMMENTS:

Recommendation: Minor Revision

Comment 1: There is much to admire about this paper – the care with which measures were chosen and tested (sample 1) and the use of multiple samples to test the hypotheses. However, I do feel some improvements are required, as detailed below.

Response: Thanks

Comment 2: The paper needs thorough overhaul in terms of language and much sleeker presentation is required – I found myself having to read some passages multiple times to understand them.

Response: The comment is very generic, however we reviewed the manuscript and made necessary changes to improve readability.

Comment 3: The hypotheses also need a lot more development. The introduction introduces a lot of concepts, is repetitive, but doesn’t explain in any detail the hypotheses. H2a to H2c in particular could be argued the other way around and make more sense the other way around - that is supervisor support lessens the combined effects of job tension and risk factors. Indeed, you state “In this context, supervisor or organizational support will not assist in thriving or a sense of vitality (Kleine, Rudolph & Zacher, 2018) to improve well-being” (p 7 line 331-333). If this is the case, then the combined effect of tension and risk factors etc will be the same regardless of the level of supervisor support. But this is not what you state in the hypotheses and not what you test. Therefore, you need a clearer and more detailed explanation of these hypotheses. It seems to me that supervisor political support can have negative as well as positive connotations and therefore could be (in JDR terms) as resource in some circumstances but a demand in others. This line of reasoning needs to be drawn out a lot more.

Response: The latter part of your comment suggests that the supervisor pollical support can have both negative and positive effect on employee well-being depending on supervisor pollical support as a resource or demand and hence hypotheses H2a to H2c can be in positive or negative directions. We agree to this suggestion, but the focus of our study is to extend the literature by highlighting that the supervisor political support will strengthen the negative cumulative effects of HHW and JT on employee well-being based on the ceiling effect of human energy theory. That is, in a lay person term the proposed hypotheses aim to test that the supervisor political support will not be significantly effective in minimizing the cumulative effects of the dimensions of health harm of work and job tension on employee well-being based on this theory. Hence, the proposed hypotheses H2a to H2c are framed to test the negative or strengthening role of supervisor political support in enhancing the negative moderation effects of HHW and JT on employee well-being. No revision is made to the manuscript in response to this comment.

Comment 4: There are also too many acronyms. For example ‘risk factors of work on psychological health (RFPH)’. It would be easier for the reader to use simpler terms.

Response: Please note that the name of dimensions of the health harm of work scale was drawn from the original article published in 2016. Hence, it is not appropriate to explain the dimensions of the health harm of work scale with simpler terms. No action is taken to revise the manuscript based on this comment.

 Comment 5: Relatedly, I’m not sure that the term ‘silo effect’ used in the paper in a few places is a term that would be familiar to readers and I wonder whether it refers to something else (e.g., main effect) – but I couldn’t be sure.

Response: The term silo effect is used thrice in this manuscript and in two occasions (page 2, line 91 and page 20, line 636) it is indicated as “silo or isolated adverse effect of work”. However, based on your comment we have now revised the term silo effect with ‘silo or isolated’ in page 7, line 335, which refers to the third mention.

Comment 6: There are also needs to be a clear definition of the term ‘ceiling effect’ early in the paper.

Response: Please refer to the definition for ceiling effect provided in the early part of the Introduction section, which is in page 2, line 51 -53. Hence, no action is taken on this comment.

Comment 7: P 2 line 84 ‘positive and negative affectivity towards working conditions of jobs’ – do you really mean trait affect – which is what ‘affectivity’ usually refers to.

Response: We have now revised page 2, line 79 and 82 to indicate that affectivity is about feelings or emotions.

Comment 8: P 3 line 173-176 ‘HHW augments the theory of harm of work about the restrictions imposed by the job on employees as stakeholders. Specifically, this research advocates that employees involved in non-work-related activities improve positive health and well-being outcomes while enhancing organizational performance’ Do you mean any non-work-related activity? Surely only positive ones (which will need to be defined on why they are positive).

Response: We have now revised the manuscript to include examples of non-work related activities that has the potential to improve positive health and well-being among employees (page 5, line 189-190).

Comment 9: I’m not sure section 2.2 is needed at all.

Response: Section 2.2 in the manuscript is essential because it sets the context for the next section 2.3 which deals with the complementary effect of health harm of work and job tension on employee well-being. No action is taken on this comment.

Comment 10: Pp 10 line 429. Correlations > .70. Table 1 indicates two correlations above .70. Could we have some clarification please?

Response: Sorry, it was a typo error and revised according.

Comment 11: P 14, line 495-496 – surely H2c is only partially accepted given it was found in only one sample.

Response: Please note that the findings revealed significant three-way interaction of side effects of work, job tension, and supervisor political support on well-being (see Table 5) only in Sample 2. Hence, H2c was accepted based on the Sample 2 data, but the findings were not consistent across the two samples (Sample 2 and Sample 3). Therefore, as indicated by the reviewer we cannot draw the inference that “H2c is partially accepted” because of the inconsistent findings across the two samples. No action is taken on this comment.